# A Power Calculation Algorithm for Single-Phase Droop-Operated-Inverters Considering Linear and Nonlinear Loads HIL-Assessed

**Jorge El Mariachet** [1,*], **Jose Matas** [1], **Helena Martín** [1], **Mingshen Li** [2], **Yajuan Guan** [2] and **Josep M. Guerrero** [2]

[1] Electric Engineering Department, Polytechnic University of Catalonia (EEBE-UPC), 08019 Barcelona, Spain; jose.matas@upc.edu (J.M.); m.helena.martin@upc.edu (H.M.)

[2] Energy Teknik Department, Aalborg University (ET-AAU), 9220 Aalborg, Denmark; msh@et.aau.dk (M.L.); ygu@et.aau.dk (Y.G.); joz@et.aau.dk (J.M.G.)

[*] Correspondence: jorge.el.mariachet@upc.edu

**Abstract:** The active and reactive powers, *P* and *Q*, are crucial variables in the parallel operation of single-phase inverters using the droop method, introducing proportional droops in the inverter output frequency and voltage amplitude references. *P* and *Q*, or *P-Q*, are calculated as the product of the inverter output voltage and its orthogonal version with the output current, respectively. However, when sharing nonlinear loads these powers, *Pav* and *Qav*, should be averaged by low-pass filters (LPFs) with a very low cut-off frequency to avoid the high distortion induced by these loads. This forces the droop method to operate at a very low dynamic velocity and degrades the system stability. Then, different solutions have been proposed in literature to increase the system velocity, but only considering linear loads. Therefore, this work presents a method to calculate *Pav* and *Qav* using second-order generalized integrators (SOGI) to face this problem with nonlinear loads. A double SOGI (DSOGI) approach is applied to filter the nonlinear load current and provide its fundamental component to the inverter, leading to a faster dynamic velocity of the droop-based load sharing capability and improving the stability. The proposed method is shown to be faster than others in the literature when considering nonlinear loads, while smoothly driving the system with low distortion levels. Simulations, hardware-in-loop (HIL) and experimental results are provided to validate this proposal.

**Keywords:** droop method; active and reactive power calculation; single-phase parallelized inverters; nonlinear loads; HIL

## 1. Introduction

The parallelization of single-phase inverters without communications between them has usually been performed using the droop method, which drives the sinusoidal references of the inverters for sharing the common loads [1–3]. The method introduces proportional droops in the inverter frequency $\omega^*$ and voltage amplitude $V^*$ references, respectively, according to the *P* and *Q* load consumed powers. These powers are usually obtained as the product of the measured output current $i_o(t)$ with the measured output voltage $v_o(t)$ and its quadrature version, $v_{o\perp}(t)$, respectively. The droop method works well for the sharing of linear loads but not for nonlinear ones, due to the nonlinear currents drawn by these loads, which highly distort *P* and *Q*, and the provided droop references. Therefore, LPFs in *P* and in *Q* with a very low cut-off frequency are used to deal with this problem. The LPFs provide the average powers consumed by the load *Pav* and *Qav* and removes the double frequency components

resulting from the product of $i_o(t)$ with $v_o(t)$ and $v_{o\perp}(t)$. However, these LPFs force the droop to run slowly, with a low dynamic response to changes in the load, worsening the system stability [4–15].

The calculation of *Pav* and *Qav* is performed by the droop-based local control algorithm of the single-phase inverters and needs of the $i_o(t)$, $v_o(t)$ and $v_{o\perp}(t)$ inverter sensed signals. The quadrature output voltage $v_{o\perp}(t)$ has been performed using different approaches applied to $v_o(t)$, such as a transport delay (TD) [15,16], an extended three-phase dq-Synchronous Reference Frame (dq-SRF) approach [17,18], and the SOGI filter [19]. Additionally, *Pav* and *Qav* have been obtained by different approaches for improving the performance of the droop method. In [20], a method based on the SOGI structure for the calculation of *Pav* and *Qav* was proposed. The method was similar to [9], and cancelled the double frequency components resulting from the products in the same manner. In this case, the dynamic response of the calculated powers was reduced by one order of magnitude. However, this work only considered the use of linear loads, and used LPFs to obtain *Pav* and *Qav* that constrained the dynamic response of the system. In [21], a method based on the discrete Fourier transform (DFT) was presented for extracting *Pav* and *Qav*. This method had the drawback of introducing a severe delay into the process, making it unsuitable for systems with abrupt load changes. In [22], a Least Mean Squares (LMS) approach was presented for obtaining the averaged powers through optimization of a cost function depending on *P* and *Q*. However, the validity of these approaches was only partial when sharing nonlinear loads, except for the last one, in which nonlinear loads were considered. In general, all of these proposals have in common the objective of trying to achieve a faster and accurate calculation of the averaged powers for enhance the droop system stability.

This paper is a natural continuation of our previous work [23] in which a DSOGI approach was introduced in the power calculation scheme proposed in [9] and [20]. The DSOGI achieves the fundamental component of the inverter output current and removes the LPFs from the scheme, which had until now been the main limitation on the performance of the droop controller. The DSOGI has an inherent trade-off relationship between its filtering capability for extracting the fundamental component and its transient response speed to changes in the input signal. This trade-off is regulated by the adjustment of the DSOGI damping factor $\xi$. The DSOGI trade-off is better than that achieved by the standard SOGI. Then, the DSOGI can reach a faster transient time response to changes in the load for similar filtering capability than the SOGI structure. In this paper, comparisons with the calculation methods of [6] and [17] when using symmetrical and non-symmetrical nonlinear loads are presented using simulations, HILplatform and experimental platform results to prove the validity of this proposal for calculating the average powers.

## 2. Materials and Methods

The scheme of a single-phase voltage source inverter (VSI) operated with the droop method when sharing a nonlinear load $Z_{NL}$ with another inverter is depicted in Figure 1. From the scheme, the main parts considered in this work can be seen: the power stage with inner controller, the pulse width modulation block (PWM), the *LC* output filter, the *vo* and *io* sensing, the *Pav* and *Qav* calculation block, and the droop generator producing the voltage reference *vref* [24,25]. In Figure 1, the scheme of the second inverter is not depicted. Rather, it is only outlined with discontinuous lines and indicated as #Inv. 2, due to the fact that this proposal is only concerned with the dynamic behavior and accuracy of the *P-Q* calculation block. For this reason, the simulations and experimental results shown from now on will only correspond to a single VSI.

In the simulations and experiments, a diode bridge rectifier (DBR) supplying a RC load is used as a nonlinear load $Z_{NL}$ (see Figure 2). At steady state, $Z_{NL}$ draws a distorted and symmetrical nonlinear current with peak levels reaching ±2.48A for the $Z_{NL}$ specified in Table 1, which induces a high distortion in *Pav* and *Qav*, as well as in $v_{ref}$. Likewise, a switch *S1* is inserted in the $Z_{NL}$ scheme (see Figure 2), allowing for step-perturbations for testing the dynamic behavior of *Pav* and *Qav* and for assessment and comparison purposes.

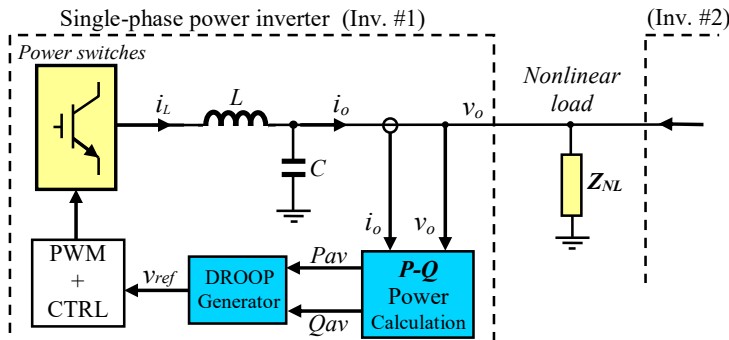

**Figure 1.** Block scheme of a single-phase voltage source inverter (VSI) (Inv. #1) sharing a nonlinear load with a second inverter (Inv. #2) and showing the *P-Q* power calculation, the Droop Generator and the PWM (pulse width modulation) control blocks.

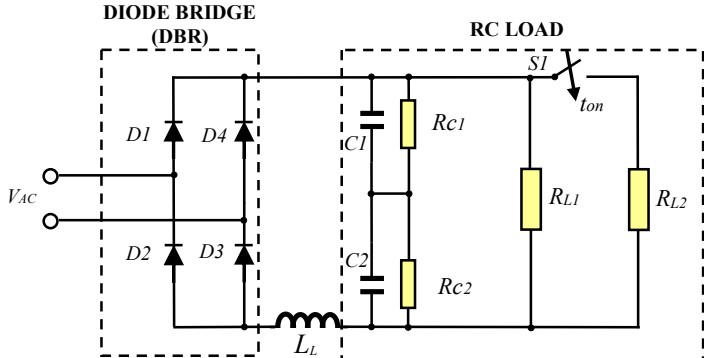

**Figure 2.** Diode-bridge rectifier nonlinear load supplying a resistive-capacitive load.

**Table 1.** Main parameters of VSI and $Z_{NL}$.

| Parameter | Value |
|---|---|
| *LC* Filter | 1.8 mH; 25 µF |
| *Switching frequency, fs* | 10 kHz |
| $Ron_{D1\text{-}D4}$ | 0.01 Ω |
| $L_L$ | 84 µH |
| C1, C2 | 470 µF |
| $R_{C1}$, $R_{C2}$ | 37 kΩ |
| $R_{L1}$, $R_{L2}$ | 960 Ω |

Simulations of the proposed system for the design and comparisons with [6,17] were performed with Matlab/Simulink/Simscape Power Systems software. Table 1 shows the main parameters of the inverter and $Z_{NL}$.

All the power calculation schemes were tuned to achieve the same power-ripple in order to obtain a fair comparison between the methods and to accurately measure the settling time of their transient responses when $Z_{NL}$ changes suddenly. The simulation results were obtained using Matlab/Simulink/Simscape Power Systems software, and were contrasted with HIL results at the inverter-based intelligent Microgrid Laboratory (iML) of the department of Energy Technology at the Aalborg University (iML-AAU) in Denmark (Figure 3).

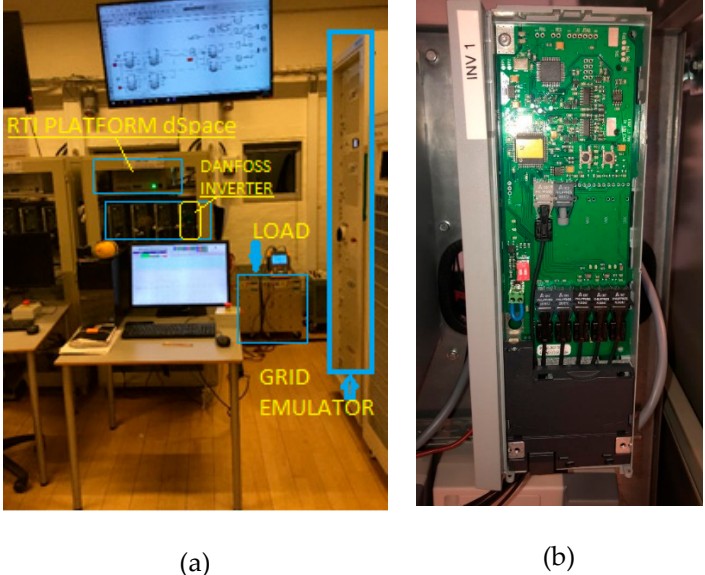

(a)　　　　　　　　　　　(b)

**Figure 3.** Experimental setup at the intelligent Microgrid Laboratory in Aalborg University (iML-AAU), Denmark: (**a**) Complete experimental setup; (**b**) detail of the Danfoss© single-phase inverter.

A second similar test was performed with another diode bridge rectifier nonlinear load drawing an asymmetrical nonlinear current in order to further test the calculation power block. This current reached a peak of 4.2 A in the positive half-cycle of the VSI and a negative one of −2.2 A in the VSI negative half-cycle after an abrupt load change. This asymmetry in the current introduced an extra distortion to the calculated powers. Experimental results for this load with a VSI inverter of the iML-AAU were also obtained. The inverter used in the iMG is a Danfoss© FC302, 2.2 kW rated, interfaced to a real-time dSPACE 1006. The algorithms for operating the VSI are developed in Matlab/Simulink software and compiled into the dSPACE. Figure 4 depicts the scheme of this experimental setup.

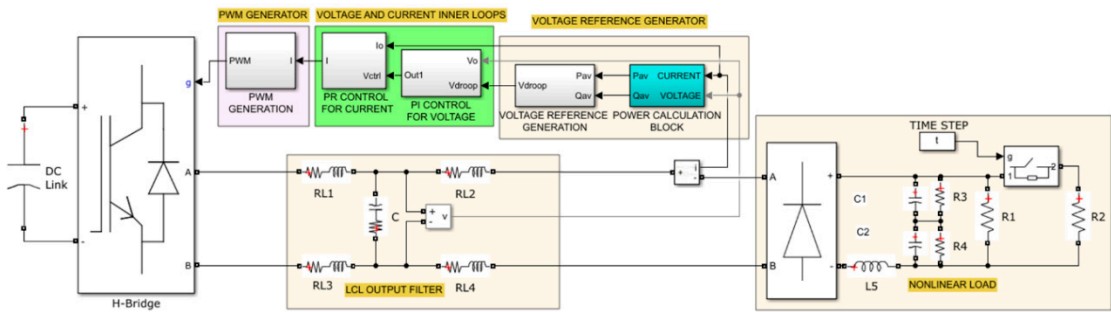

**Figure 4.** Scheme of the VSI experimental setup implemented at the iMG_AAU.

### 2.1. Droop Principle-Based Control Scheme

The power calculation block of Figure 1 provides the frequency and voltage amplitude references $\omega^*$ and $V^*$, respectively, as can be seen in the following equations:

$$\omega^* = \omega_n - m \cdot P_{av} \tag{1}$$

$$V^* = V_n - n \cdot Q_{av} \tag{2}$$

where $m$ and $n$ are the droop coefficients and $\omega_n$ and $V_n$ are the nominal frequency and voltage amplitude, respectively. These references are used to generate the sinusoidal voltage reference $v_{ref}$ for the inverter inner control loops:

$$v_{ref} = V^* \cdot sin(\omega^* \cdot t) \tag{3}$$

Assuming that the output voltage and current of the inverter are [15]

$$v_o(t) = V \cdot sin(\omega_o \cdot t) \tag{4}$$

$$i_o(t) = I \cdot sin(\omega_o \cdot t - \varphi_o) \tag{5}$$

where $V$ and $I$ are the voltage and current amplitudes, $\omega_o$ is the fundamental frequency and $\varphi_o$ is the phase angle between $v_o(t)$ and $i_o(t)$, the quadrature voltage, with a $-\pi/2$ delay, is defined as

$$v_{o\perp}(t) = V \cdot sin\left(\omega_o \cdot t - \frac{\pi}{2}\right) \tag{6}$$

Therefore, the instantaneous active and reactive powers can be formulated as

$$p_i(t) = v_o(t) \cdot i_o(t) = \frac{VI}{2} \cdot [cos\varphi_o - cos(2\omega_o \cdot t - \varphi_o)] = P_{av} + \widetilde{p} \tag{7}$$

$$q_i(t) = v_{o\perp}(t) \cdot i_o(t) = \frac{VI}{2} \cdot [sin\varphi_o - sin(2\omega_o \cdot t - \varphi_o)] = Q_{av} + \widetilde{q} \tag{8}$$

where $\widetilde{p}$ and $\widetilde{q}$ are the oscillating components that pulsate at twice the fundamental frequency $\omega_o$. These equations reveal that for a linear load that draws a sinusoidal current, the instantaneous powers oscillate around the averaged powers $P_{av}$ and $Q_{av}$. However, if the load is nonlinear, the instantaneous powers are going to be highly distorted by harmonics, for which the nonlinear current can be expressed as:

$$i_o(t) = I_0 \cdot sin(\omega_o t - \varphi_o) + \sum_{h=2}^{N} I_h \cdot sin(h \cdot \omega_o \cdot t - \varphi_h) \tag{9}$$

leading to

$$p_i(t) = P_{av} + \widetilde{p} + v_o(t) \cdot \sum_{h=2}^{N} I_h \cdot sin(h \cdot \omega_o \cdot t - \varphi_h) \tag{10}$$

$$q_i(t) = Q_{av} + \widetilde{q} + v_{o\perp}(t) \cdot \sum_{h=2}^{N} I_h \cdot sin(h \cdot \omega_o \cdot t - \varphi_h) \tag{11}$$

where the subscript $h$ represents the harmonic order, $N$ is the maximum considered value for $h$ and $I_h$, $h \cdot \omega_o$ and $\varphi_h$ are the amplitude, the frequency and the phase-shift of the current harmonic components, respectively. As can be seen in Equations (10) and (11), $p_i(t)$ and $q_i(t)$ contain higher harmonic order components, in addition to the DC $P_{av}$ and $Q_{av}$ and the double frequency components $\widetilde{p}$ and $\widetilde{q}$ that were already present in the linear case. Then, for a nonlinear $Z_{NL}$, Equations (1) and (2) can be expressed as follows when they are calculated employing the instantaneous powers in Equations (10) and (11):

$$\omega^*(t) = \omega_n - m \cdot \left( P_{av} + \widetilde{p} + v_o(t) \cdot \sum_{h=2}^{N} I_h \cdot sin(h \cdot \omega_o \cdot t - \varphi_h) \right) \tag{12}$$

$$V^*(t) = V_n - n \cdot \left( Q_{av} + \widetilde{q} + v_{o\perp}(t) \cdot \sum_{h=2}^{N} I_h \cdot sin(h \cdot \omega_o . t - \varphi_h) \right) \tag{13}$$

and the voltage reference is

$$v_{ref}(t) = V^*(t) \cdot sin(\omega^*(t) \cdot t) \tag{14}$$

### 2.2. Conventional P-Q Calculation Schemes

Figure 5 shows the conventional *P-Q* calculation scheme for obtaining $P_{av}$ and $Q_{av}$, in which the instantaneous powers $p_i(t)$ and $q_i(t)$ are obtained as the products of $i_o(t)$ with $v_o(t)$ and $v_{o\perp}(t)$, respectively [6]. The quadrature voltage $v_{o\perp}(t)$ is obtained by delaying $v_o(t)$ by $\pi/2$, and $P_{av}$ and $Q_{av}$ by using LPFs with a low cut-off frequency value *fc*, to filter the multiple frequency components in the instantaneous powers [18]. When sharing linear loads, the value of *fc* is usually set to one or two orders of magnitude lower than the inverter fundamental operating frequency [22,23], which determines the transient dynamic performance of the system. However, in the case of nonlinear loads, the value of *fc* should be further reduced (usually to less than 1 Hz), to avoid strong distortions in the inverter output current and in the instantaneous powers [24]. Conversely, the distortion in the current induces excessive ripple in the averaged powers, which in turn is translated to the droop references $\omega^*(t)$ and $V^*(t)$, and then to $v_{ref}(t)$, causing bad operation of the system. Nevertheless, this bandwidth reduction slows down the transient dynamic behavior of the system. The transfer functions of the averaged $P_{av}$ and $Q_{av}$ are shown in Appendix A.

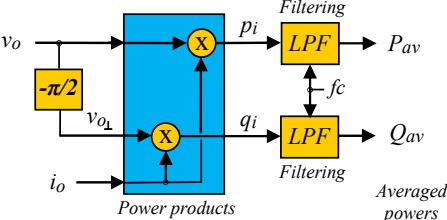

**Figure 5.** Block diagram of conventional *P-Q* power calculation providing the averaged powers $P_{av}$ and $Q_{av}$.

### 2.3. The SOGI and DSOGI Approach

A SOGI is a special linear filter with one input, $v_{in}(t)$, and two outputs, $v_d(t)$ and $v_q(t)$, one in phase and the other delayed $\pi/2$ with respect to the input, respectively, see Figure 6.

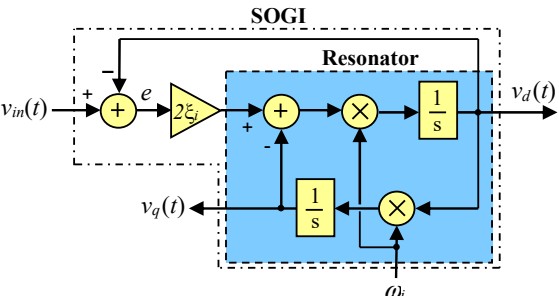

**Figure 6.** Block diagram of the SOGI (second-order generalized integrator) filter.

The outputs of the SOGI filter, $v_d(t)$ and $v_q(t)$, have the following band-pass filter (BPF) and LPF transfer function relationships regarding to the input:

$$H_d(s) = \frac{v_d(s)}{v_{in}(s)} = \frac{2\xi_i\omega_i \cdot s}{s^2 + 2\xi_i\omega_i \cdot s + \omega_i^2} H_d(s) = \frac{v_d(s)}{v_{in}(s)} = \frac{2\xi_i\omega_i \cdot s}{s^2 + 2\xi_i\omega_i \cdot s + \omega_i^2} \tag{15}$$

$$H_q(s) = \frac{v_q(s)}{v_{in}(s)} = \frac{2\xi_i\omega_i^2}{s^2 + 2\xi_i\omega_i \cdot s + \omega_i^2} H_q(s) = \frac{v_q(s)}{v_{in}(s)} = \frac{2\xi_i\omega_i^2}{s^2 + 2\xi_i\omega_i \cdot s + \omega_i^2} \tag{16}$$

where $\xi_i$ is the filter damping factor and $\omega_i$ is the tuning center frequency. In addition, a DSOGI is a four-order filter that consist in the cascade connection of two SOGI filters, as seen in Figure 7 [26]:

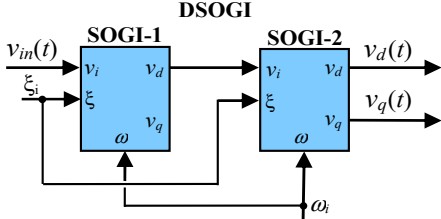

**Figure 7.** Block diagram of a DSOGI (double SOGI) filter as the cascading of two individual SOGIs.

The transfer functions of the DSOGI with respect to the input [26]:

$$H'_d(s) = \left( \frac{2\xi_i\omega_i \cdot s}{s^2 + 2\xi_i\omega_i \cdot s + \omega_i^2} \right)^2 \tag{17}$$

$$H'_q(s) = \frac{4\xi_i^2\omega_i^3 \cdot s}{\left( s^2 + 2\xi_i\omega_i \cdot s + \omega_i^2 \right)^2} \tag{18}$$

When $\omega_i$ is tuned to match $\omega_o$, and for harmonic components higher than the fundamental ($h \gg 1$), the gain of the BPF characteristic in Equation (17) can be simplified to:

$$\left| H'_d(s) \right| = \left( \frac{2\xi_i h}{\sqrt{(1 - h^2)^2 + (2\xi_i h)^2}} \right)^2 \approx \left( \frac{2\xi_i}{h} \right)^2 \left| H'_d(s) \right| = \left( \frac{2\xi_i h}{\sqrt{(1 - h^2)^2 + (2\xi_i h)^2}} \right)^2 \approx \left( \frac{2\xi_i}{h} \right)^2 \tag{19}$$

On the other hand, the frequency and damping factor are the parameters that determine the settling time $t_s$ (2% criterion) of the transient response of the BPF in Equation (15) for a step input:

$$t_s \approx \frac{4}{\xi_i \cdot \omega_i} \tag{20}$$

The SOGI filter has an inherent trade-off relationship between bandwidth (rejection capability to harmonics) and settling time response, Equation (20). This trade-off cannot be overcome, and relies on the damping factor parameter $\xi_i$. However, in [26], it was shown that the DSOGI has a better trade-off than the SOGI and can achieve a faster transient response when it is designed to have the same bandwidth behavior. In this paper, this characteristic is used to achieve a faster response when extracting the fundamental component of the nonlinear load current and thereby to improve the droop transient response to load changes.

*2.4. Advanced P-Q Calculation Scheme*

Figure 8 shows a *P-Q* calculation method based on the proposed scheme in [17] for accelerating the computation of $P_{av}$ and $Q_{av}$. In this figure, the *SOGI1* and *SOGI2* blocks are used for extracting the double frequency pulsating power components $\widetilde{p}$ and $\widetilde{q}$, which are then removed from $p_i$ and $q_i$. These SOGIs are both tuned at $\omega_i = 2\omega_o$ and $\xi_1 = \xi_2 = 1$. The LPFs are used for improved filtering and for providing the averaged powers $P_{av}$ and $Q_{av}$, by attenuating the higher harmonics components reported in Equations (10) and (11). This figure does not show the method for generating the $\pi/2$ delay, since it is not mentioned in [17]. Therefore, another SOGI, *SOGI0*, tuned at $\omega_o$ and $\xi_0 = \xi_v = 0.707$, is used for generating this delay, as shown in Figure 9, for avoiding the delay issues reported in [15,16].

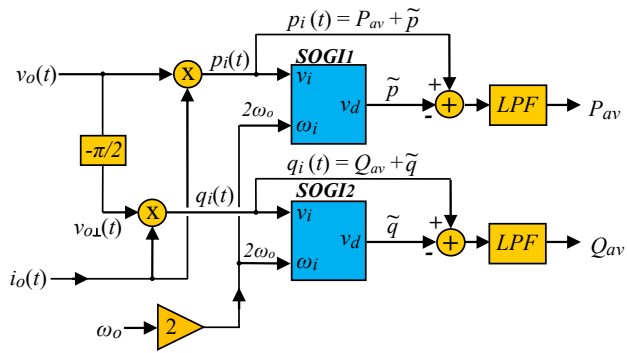

**Figure 8.** Scheme for the calculation of $P_{av}$ and $Q_{av}$ proposed in [20].

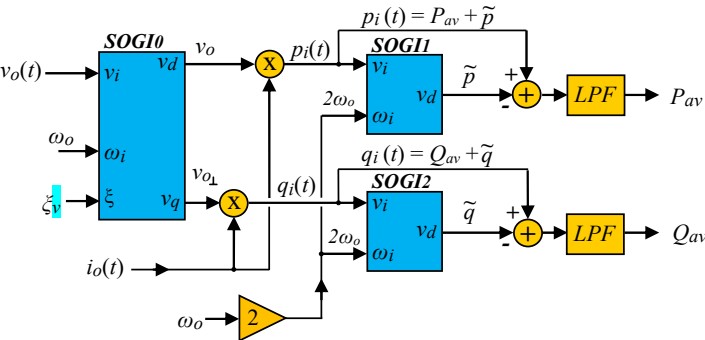

**Figure 9.** Addition to Figure 8 of a new SOGI for archiving the quadrature component of the output voltage $v_{o\perp}(t)$.

The transfer functions for the calculated averaged active and reactive power are included in Appendix A, with the numbers (24) and (25). Please note that the current is not filtered.

Figure 10 shows the simulation results using the P-Q algorithms of the schemes in Figures 5 and 9 when sharing a linear load that produces a current perturbation from peak 2A to peak 4A at a time of 3 s.

For the sake of simplicity, Figure 10 only shows the dynamic behavior of $P_{av}$, obviating the representation of $Q_{av}$. The dynamics of $P_{av}$ obtained with the advanced scheme in Figure 9 (hereinafter called $P_{adv}$) are compared with those obtained for $P_{av}$ by the conventional droop method depicted in Figure 5 (hereinafter called $P_{droop}$). The cut-off frequency of the LPFs in Figure 9 was designed to be $fc = 3.7$ Hz, whereas it was set to $fc = 0.37$Hz for the scheme in Figure 5. As shown in Figure 10, the double frequency component $\widetilde{p}$ was removed from $P_{adv}$. Likewise, the higher cut-off frequency of its LPFs results in much faster dynamics than those of $P_{droop}$. Also, the ripple corresponding to the double frequency component $\widetilde{p}$, which is not completely filtered by the LPFs in Figure 5, can be observed in $P_{droop}$. These results are compatible with those reported in [21]. However, the good dynamic behavior depicted in Figure 10 for $P_{adv}$ vanishes when a nonlinear load is used, as is shown in Figure 11. In this case, the nonlinear load is a DBR that draws a peak current of ±2.48 A and suffers a perturbation that pushes the peak to ±4.15 A. The simulation parameters are listed in Table 2.

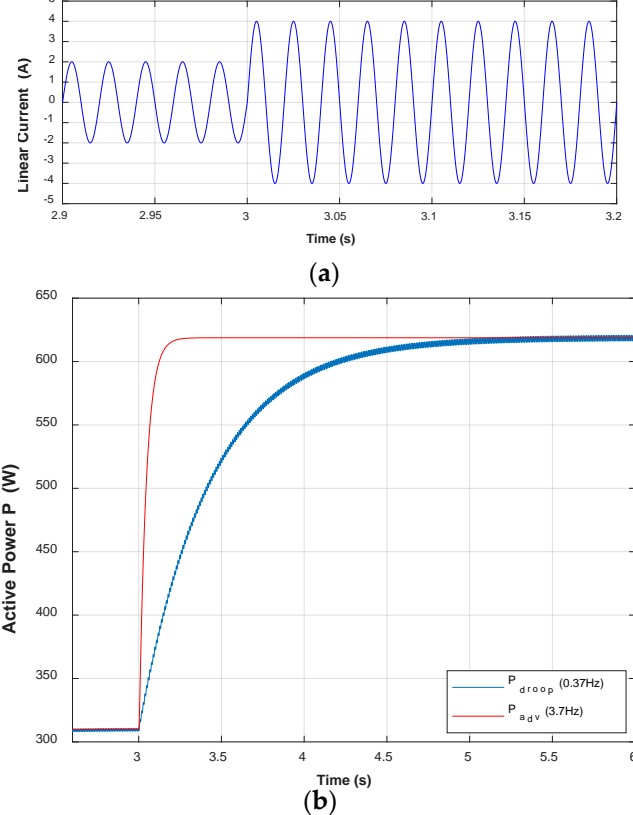

**Figure 10.** Simulation of the load current and $P_{av}$ transient responses for a linear load current perturbation from peak 2A to peak 4A at 3 s: (**a**) detail of the load current perturbation, (**b**) comparison between $P_{av}$ obtained with Figure 9 scheme ($P_{adv}$) and $P_{av}$ obtained with Figure 5 scheme ($P_{droop}$).

**Table 2.** Simulation parameters for Figure 10.

| Parameter | Value |
|---|---|
| $V_n$ | 311 V |
| $\omega_o$ | $2\pi50$ rad/s |
| *DBR RESISTOR LOAD* FOR T < 3S | 950 Ω |
| *DBR RESISTOR LOAD* FOR T > 3S | 471.8 Ω |
| THD $i_o(t)$ | 215% |
| $\xi_0$ | 0.7 |
| $\xi_1 = \xi_2$ | 1 |

As shown in Figure 11, the dynamics of the method proposed in [20] worsen using a nonlinear load, similarly to [26]. Thus, in the presence of a nonlinear load, the method has excessive steady state ripple that corrupts the calculated powers, opposite to what is stated in [20]. To reduce the ripple, the filtering capabilities of the LPFs in Figure 9 can be improved by reducing *fc*. By decreasing *fc* from 3.7 Hz to 1.1 Hz, the ripple of $P_{adv}$ matches that of $P_{droop}$, as can be seen in Figure 12. Although the advanced method of Figure 9 calculates faster $P_{av}$ than the conventional droop controller, it presents less effectivity than initially argued. It can be clearly seen in Figures 11 and 12 that there is a trade-off between the filtering capability of the $P_{av}$ calculation scheme and the transient speed of its dynamical response. Note also that there is a positive offset in the calculated $P_{av}$ at steady state, since the mean value of $P_{adv}$ is slightly higher than that of $P_{droop}$ (see Figure 11). Comparing the transfer functions for the P-Q calculation by the conventional, Equations (22) and (23), or by the advanced method, Equations (24) and (25), it can be seen the filtering capabilities of each algorithm. Please note that the

magnitude of the ripple of $P_{adv}$ forces the system's dynamic response to slow down by reducing more the cut-off frequency.

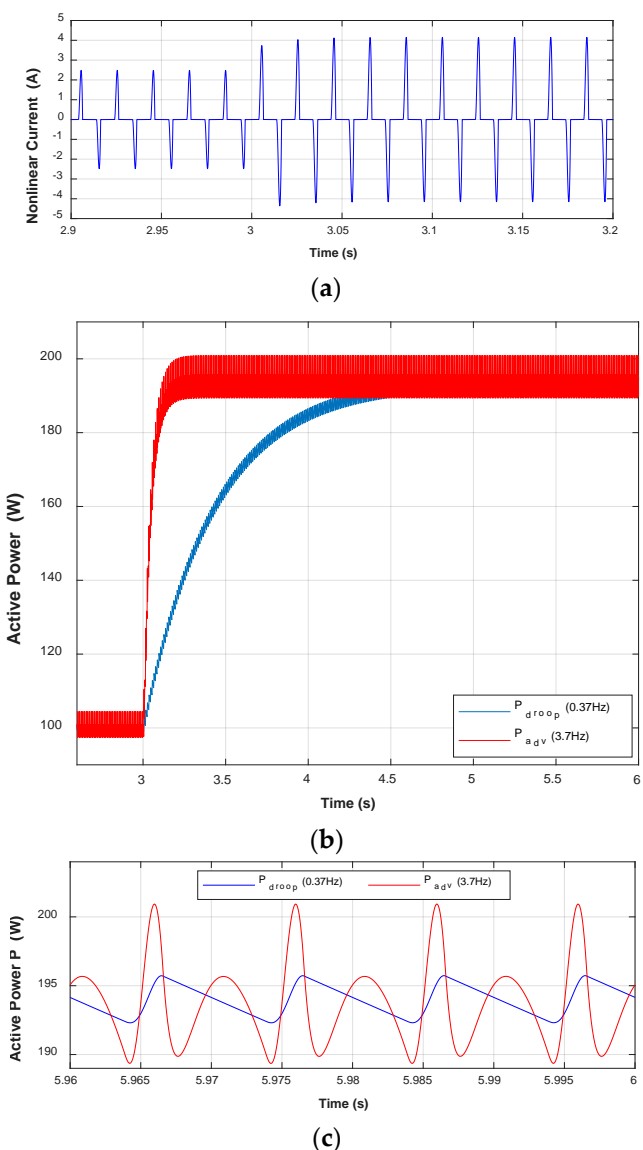

**Figure 11.** Simulation of the load current and $P_{av}$ transient responses for a nonlinear Diode Bridge Rectifier (DBR) load current perturbation from a peak at 2.48 A to a peak at 4.15 A at 3 s: (**a**) detail of the distorted load current perturbation, (**b**) $P_{adv}$ and $P_{droop}$, (**c**) detail of $P_{adv}$ and $P_{droop}$ ripples.

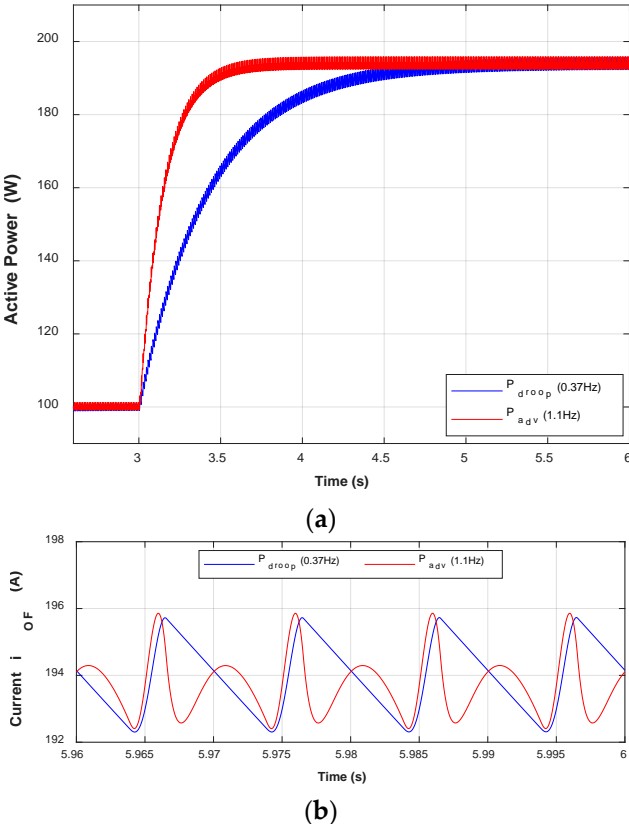

**Figure 12.** Simulation of the $P_{av}$ transient responses for a nonlinear DBR (diode bridge rectifier) load current perturbation from a peak at 2.48 A to a peak at 4.15 A at 3 s: (**a**) $P_{adv}$ (with reduced $fc = 1.1$ Hz) and $P_{droop}$, (**b**) detail of $P_{adv}$ and $P_{droop}$ ripples.

### 2.5. Proposed P-Q Calculation Scheme

The scheme of the proposed *P-Q* calculation is shown in Figure 13. A DSOGI approach is applied to $i_o(t)$ in order to extract the fundamental component $i_{OF}$, see Equation (28) in Appendix A. There, the DSOGI composed of the *SOGI3* and *SOGI4* blocks filters the distorting high-order harmonics of $i_o(t)$ by means of its higher BPF capability and avoids coping with a highly distorted current signal. Consequently, the instantaneous powers $p_i(t)$ and $q_i(t)$ are obtained as the product of the in-phase $v_o(t)$ or the quadrature $v_{o\perp}(t)$ voltages with the fundamental output current $i_{OF}$, respectively. This produces a result with only double frequency components and without third or higher order harmonics.

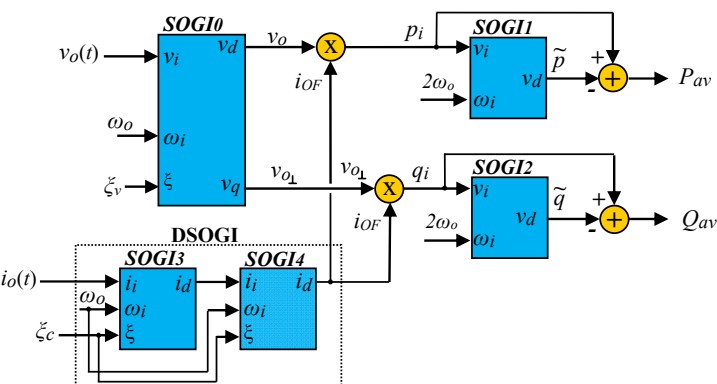

**Figure 13.** Scheme for the proposed *P-Q* calculation based on a DSOGI approach.

Later, the *SOGI1* and *SOGI2* blocks were used as in Figure 9 for removing only the double frequency components with the help of the subtracting blocks. Therefore, the LPFs can now be removed from the scheme, since they are no longer necessary. The transfer function of the proposed scheme in Figure 13 is shown in Appendix A as Equations (26) and (27). This overcomes the main limitation of previous schemes and further accelerates the dynamic response of $P_{av}$ and $Q_{av}$. To distinguish $P_{av}$ and $Q_{av}$ obtained with this proposed scheme, in the following they will be referred to as $P_{DSOGI}$ and $Q_{DSOGI}$.

In this case, because a DSOGI is used for filtering the current in Equation (28), and considering that the center frequencies provided by the droop method $\omega^*$ vary in a small range around the nominal frequency $\omega_n$, the transient response speed is determined mainly by the DSOGI transient response, which is related to $\xi_c$; see [26]. The damping factors of the DSOGI are here tuned to produce a power ripple identical in amplitude to that of the conventional droop controller, which is achieved for $\xi_c = \xi_3 = \xi_4 = 0.129$.

Figure 14 shows the simulation results of the same nonlinear DBR load drawing a peak current of ±2.48 A and suffering a perturbation that pushes the peak to ±4.15 A. It can be observed that the proposed method is faster for calculating $P_{av}$ than the previously considered methods.

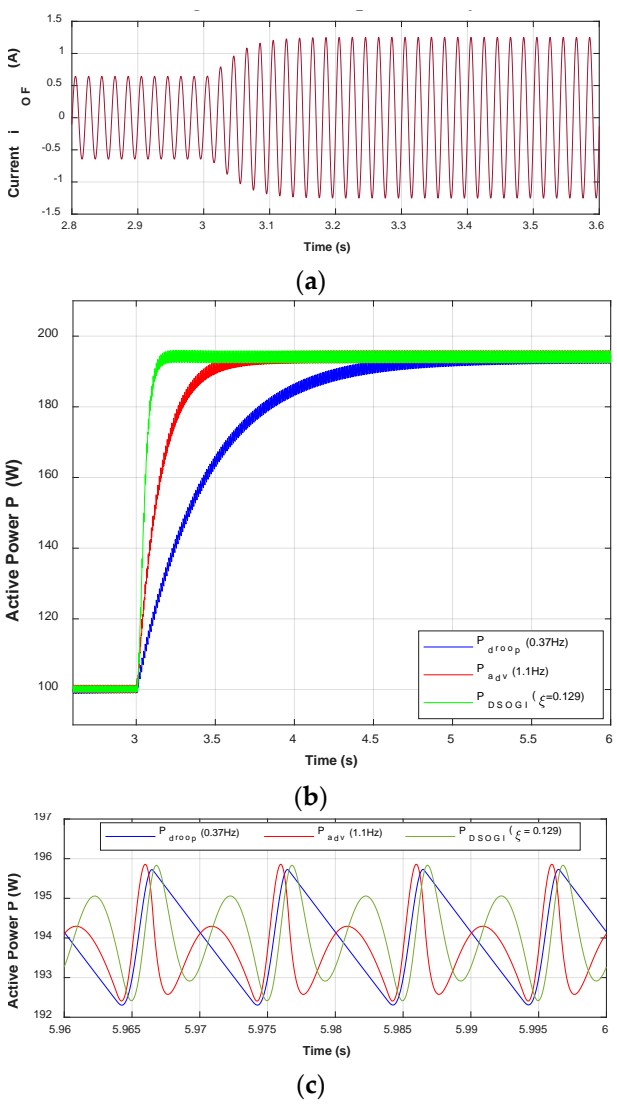

**Figure 14.** Simulation of $i_{OF}$ and $P_{av}$ transient responses for a nonlinear DBR load current perturbation from a peak of 2.48 A to 4.15 A at 3 s: (**a**) detail of the $i_{OF}$ perturbation, (**b**) $P_{DSOGI}$, $P_{adv}$ and $P_{droop}$, (**c**) detail of $P_{DSOGI}$, $P_{adv}$ and $P_{droop}$ ripples.

Figure 15 depicts the simulation results using a $Z_{NL}$ drawing an asymmetrical current, reaching a positive peak of 5.21 A and a negative peak of −3.07 A after the perturbation. In this case, the asymmetry in the nonlinear current (see Figure 15a), introduces further distortion into the calculated $P_{av}$, showing higher ripple at steady state (see Figure 15c).

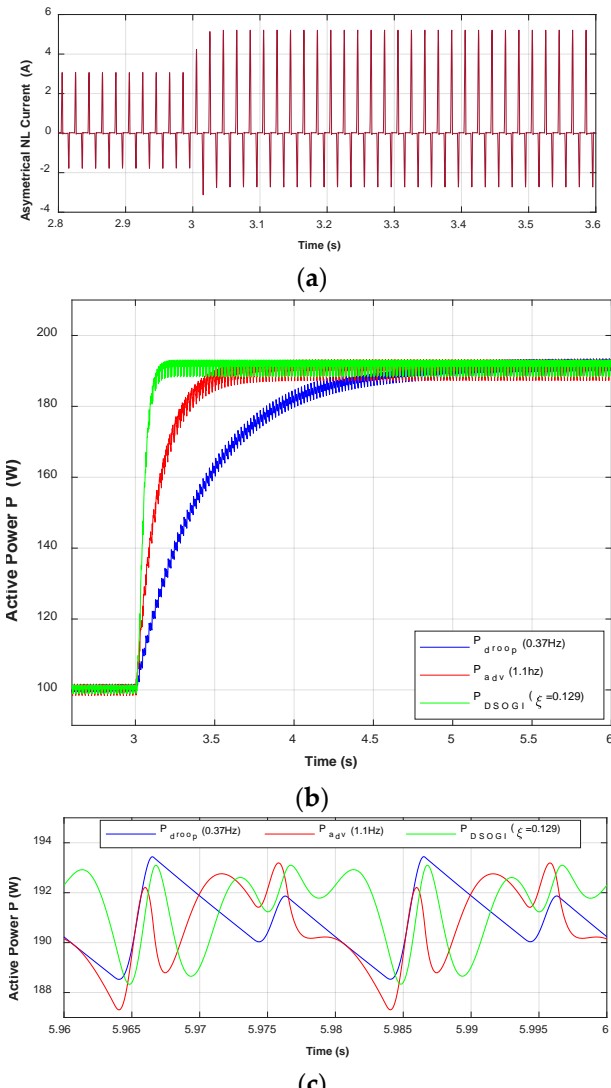

**Figure 15.** Simulation of the load current and $P_{av}$ transient responses for an asymmetrical nonlinear DBR load current with a positive peak of 5.21 A and a negative peak of −3.07 A after the perturbation at 3 s: (**a**) detail of the asymmetrical load current perturbation, (**b**) $P_{DSOGI}$, $P_{adv}$ and $P_{droop}$, (**c**) detail of $P_{DSOGI}$, $P_{adv}$ and $P_{droop}$ ripples.

Please note that unlike $P_{adv}$, $P_{DSOGI}$ does not exhibit a positive offset error in steady state. This means that the proposed method is also more accurate than that in [20]. Table 3 contains the measured settling time for the $P_{av}$ transient responses depicted in Figure 15, which shows that the $P_{DSOGI}$ calculated with the proposed method implies a 60.00% and a 79.69% reduction in the response time regarding $P_{adv}$ and $P_{droop}$, respectively.

**Table 3.** Settling time simulation measurements from Figure 15.

| Parameter | Value |
|-----------|-------|
| $P_{DROOP}$ | 325 ms |
| $P_{ADV}$ | 165 ms |
| $P_{DSOGI}$ | 66 ms |

## 3. Experimental Results

Hardware in Loop, as well as experimental results with an VSI inverter of the iML-AAU, were obtained. The HIL setup consisted of a dSpace 1006© digital Real-Time Interface platform that carried out the processor-based simulations with parameters configured by the dSpace Configuration Desk © tool. The setup supports models based on the physical modelling libraries of electrical plants designed by Matlab/Simulink/SimPowerSystems©. These models are integrated into the dSpace tool chain and are used for building the electrical system under test along with the Electronic Central Unit, ECU, of the dSpace. The control algorithms described in this paper were loaded and executed in real time in the ECU. In this case, the H-Bridge Inverter, the LCL filter and the Nonlinear Load depicted in Figure 4 are the electrical system implemented in the ECU under the HIL test. The experimental setup consisted of an inverter Danfoss© FC302, 2.2 kW rated, interfaced to the real-time dSPACE 1006 digital platform. The algorithms for operating the VSI were developed in Matlab/Simulink software and compiled in the dSPACE multiprocessor core, with its parameters configured and controlled through the Configuration-Desk software. The sampling frequency for the system was 10 kHz, which was the switching frequency for the VSI. A third-order method for discretizing the SOGI and DSOGI algorithms was employed, whereby the integrator $\frac{1}{s}$ was approximated as:

$$\frac{T_s}{12} \cdot \frac{5z^{-3} - 16z^{-2} + 23z^{-1}}{1 - z^{-1}} \tag{21}$$

with *Ts* being the sample time of 100 μs, which is consistent with the 10 kHz frequency. The experimental setup parameters for the LCL filter and for the nonlinear load are listed in Table 4.

**Table 4.** Main parameters of VSI and $Z_{NL}$.

| Parameter | Value |
|-----------|-------|
| *RL1 = RL2 = RL3 = RL4* | 1.8 mH; 0.01 Ω |
| *RC branch* | 25 μF; 1 Ω |
| *Switching frequency, fs* | 10 kHz |
| $Ron_{D1\ and\ D4}$/ $Ron_{D2\ and\ D3}$ | 0.01 Ω/ 1 Ω |
| $L_L$ | 84 μH |
| *C1 = C2/ Rc1 = Rc2* | 470 μF/37 kΩ |
| $R_{L1}$, $R_{L2}$ | 960 Ω |

In this section, HIL results are shown first for a $Z_{NL}$ drawing a symmetrical current, and then the results for a VSI with a $Z_{NL}$ drawing an asymmetrical current.

Figure 16 depicts the HIL $P_{av}$ results for a $Z_{NL}$ drawing a symmetrical current with peak values transitioning from ±2.48 A to ±4.15 A after a step perturbation at 1.28 s. The dynamics of $P_{av}$ obtained with the different considered methods are represented in green for $P_{droop}$, in blue for $P_{adv}$ and in red for $P_{DSOGI}$.

Figure 17 shows a detail of the $P_{av}$ ripple waveforms at steady state, evidencing the different nonlinearities resulting from each power calculation method. There are some DC errors in the calculated powers that can mainly be attributed to the discretization method in Equation (21) and to the sampling frequency. This phenomenon was not evidenced in the simulations shown in Figure 15c, but is reflected in the HIL results.

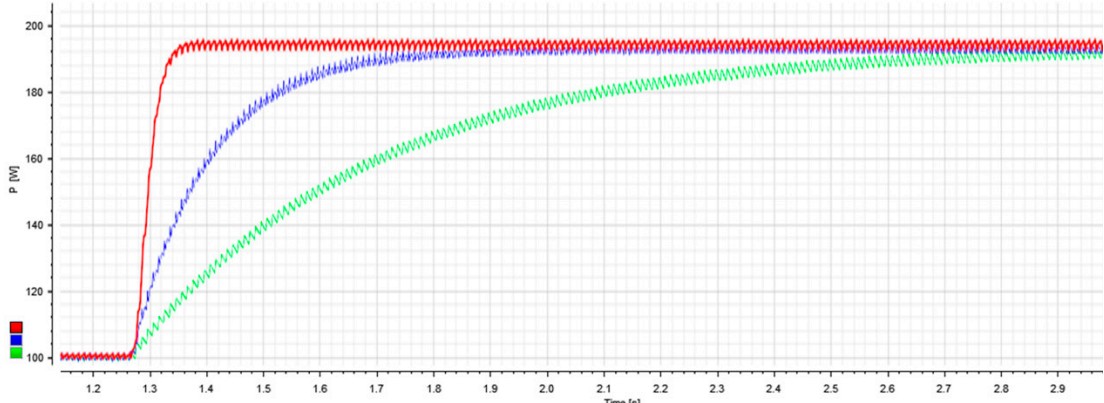

**Figure 16.** HIL $P_{av}$ results for a $Z_{NL}$ drawing a symmetrical current with peak values from ±2.48 A to ±4.15 A after a step perturbation at 1.28 s ($P_{droop}$ in green, $P_{adv}$ in blue, $P_{DSOGI}$ in red).

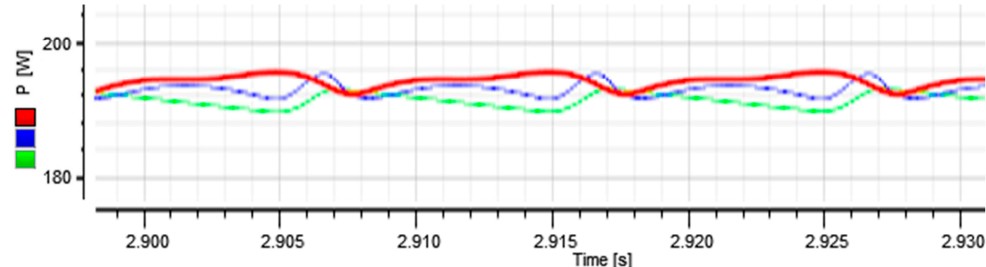

**Figure 17.** Detail of HIL $P_{av}$ ripples at steady-state from Figure 14 ($P_{droop}$ in green, $P_{adv}$ in blue, $P_{DSOGI}$ in red).

Figure 18 shows the output voltage $v_o(t)$ waveform of the VSI (Figure 18a), the current waveform drawn for the DBR load and used for obtaining the VSI results (Figure 18b) and the fundamental current component $i_{OF}$ extracted by the DSOGI (Figure 18c). Note the current asymmetry, with positive peaks that reach 4.2 A and negative ones that reach −2.8 A. Note how a fundamental current component t of 1.5 A amplitude has been achieved.

Finally, Figure 19 depicts the $P_{av}$ experimental VSI results with the DBR load when a perturbation is applied at 8.9 s. Note in this figure that the results are coherent with those obtained by HIL, as shown in Figure 16, and by the simulations in Figure 15.

Table 5 presents the settling time of $P_{av}$ obtained with all the considered P-Q calculation methods, as measured in Figure 15 for Matlab/Simulink® simulations results, in Figure 16 for the HIL results, and in Figure 19 for the experimental VSI results. In this table, the achieved percentage of settling time reduction with respect to the conventional droop method is also indicated. Note how the proposed method achieves a faster response. The transfer functions for the active and reactive averaged powers of the droop, advanced and DSOGI methods are shown in Appendix A.

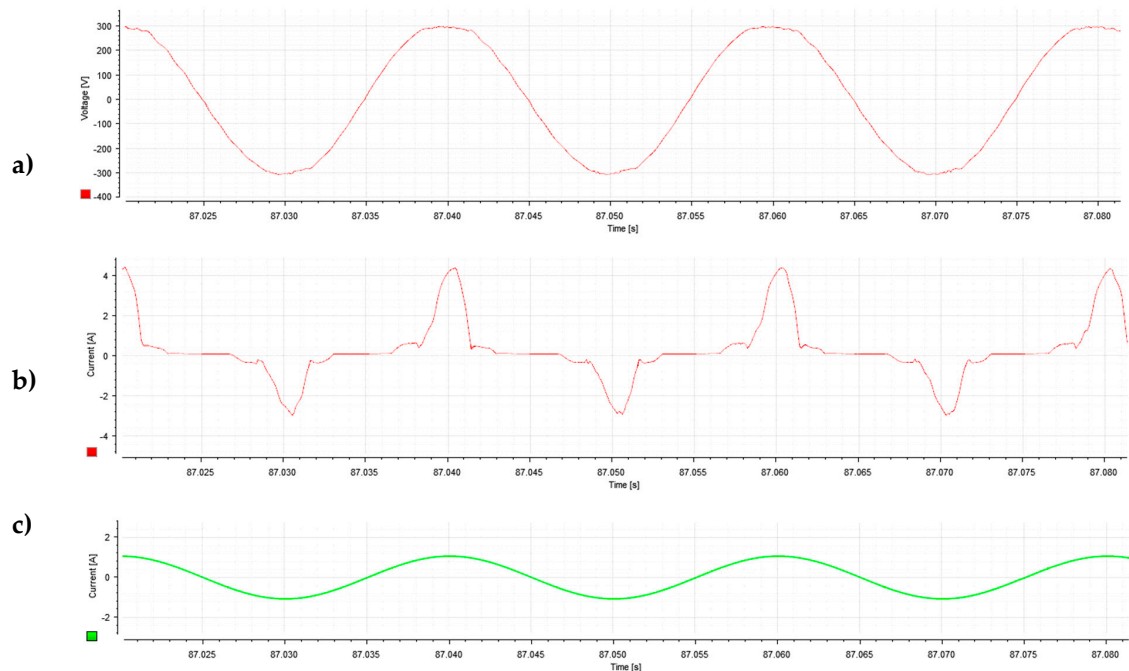

**Figure 18.** VSI output voltage and asymmetrical current waveforms drawn by the nonlinear DBR load used to obtain the VSI results. (**a**) Output voltage, $v_o(t)$; (**b**) asymmetrical nonlinear current; (**c**) fundamental component $i_{OF}$ extracted by the DSOGI.

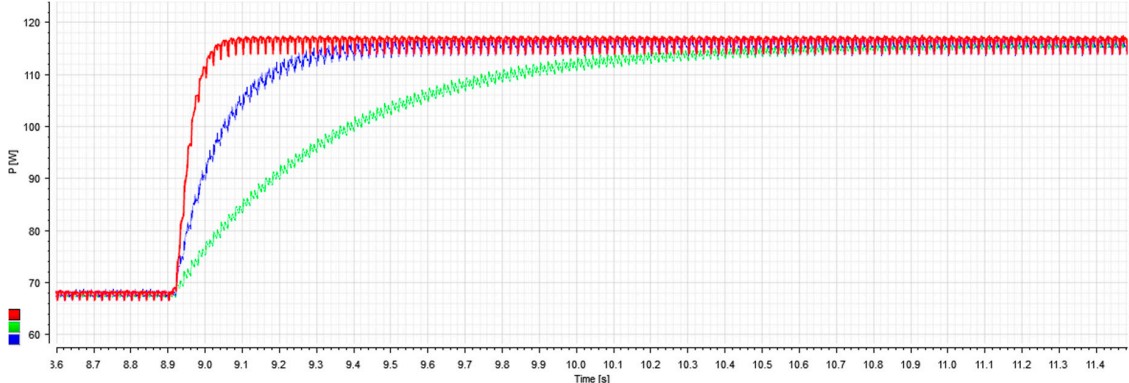

**Figure 19.** VSI experimental result of the $P_{droop}$ transient response for the nonlinear DBR load current perturbation in Figure 18 ($P_{droop}$ in green, $P_{adv}$ in blue, $P_{DSOGI}$ in red).

**Table 5.** Experimental results for the VSI setup.

| P-Q Algorithm | Settling Time Matlab (ms)/(%reduction) | Settling Time HIL (ms)/(%reduction) | Settling Time VSI (ms) /(%reduction) |
|---|---|---|---|
| Conventional, $P_{droop}$ | 760 / (—) | 780 / (—) | 930 (—) |
| Advanced, $P_{adv}$ | 310 / (59%) | 330 / (58%) | 360 (61%) |
| Proposed, $P_{DSOGI}$ | 130 / (83%) | 140 / (82%) | 180 (80%) |

## 4. Discussion

In this work, a *P-Q* calculation method was proposed for single-phase inverters with the purpose of improving the speed and accuracy of the power calculation when they are sharing linear and nonlinear loads. The dynamic response of the power calculation used in the conventional droop method and in another advanced method was first analyzed to show their limitations in speed and accuracy

when sharing a DBR-type nonlinear load. For this reason, a novel calculation method was proposed, bearing in mind the dynamic transient response velocity of the system to sudden perturbations in the shared load. The method was studied and compared with previous ones by obtaining results from Matlab®-based simulations, the HIL platform, and a VSI experimental setup. These results show how the proposed method achieves, under the same distorting conditions, a faster calculation settling time with a measured time reduction over the conventional droop method that arrives around 80%, which is higher than the achieved by the advanced method. This improvement supposes an enhancement in the droop speed operation under linear and nonlinear load conditions that leads to a better dynamic performance of the system when parallelizing inverters or microgrid operation in islanded mode.

The aim of this work was to identify what the main limitation of the power calculation methods was for sharing linear and nonlinear loads, and to propose a new approach based on the DSOGI scheme. This issue can be further investigated by using the SOGI approach and other ones for obtaining the fundamental component of the distorted current signals with less effort, faster, and best quality. This would help in the operation and stability of the inverters when sharing linear and nonlinear loads.

**Author Contributions:** Investigation, simulation results and writing—original draft, J.E.M. and J.M.; methodology and formal analysis, J.E.M., J.M., H.M. and J.M.G.; writing—review and editing, J.E.M., J.M., H.M.; writing—response to reviewers, J.M. and H.M.; software design, J.E.M., J.M. and M.L.; laboratory setup, HIL and experimental results, J.E.M., M.L., and Y.G.; data curation, J.E.M., J.M. and M.L.; laboratory organization and supervision, J.M.G.

**Funding:** J. M. Guerrero was funded by a Villum Investigator grant (no. 25920) from The Villum Fonden.

**Acknowledgments:** Thanks to Alexander Micallef, University of Malta, for his crucial support during the experimental campaign. Thanks also to the E3PACS laboratory staff from UPC, that provided technical support during the redaction of this work.

**Conflicts of Interest:** The authors declare no conflict of interest.

## Appendix A

The transfer functions of the active and reactive averaged powers regarding the instantaneous powers of the methods described here are as follows:

$$P_{droop}(s) = \left(\frac{\omega_c}{s + \omega_c}\right) p_i(s) \tag{A1}$$

$$Q_{droop}(s) = \left(\frac{\omega_c}{s + \omega_c}\right) q_i(s) \tag{A2}$$

$$P_{adv}(s) = \left(\frac{s^2 + (2\omega_o)^2}{s^2 + 2(2\omega_o) \cdot s + (2\omega_o)^2}\right)\left(\frac{\omega_c}{s + \omega_c}\right) p_i(s) \tag{A3}$$

$$Q_{adv}(s) = \left(\frac{s^2 + (2\omega_o)^2}{s^2 + 2(2\omega_o) \cdot s + (2\omega_o)^2}\right)\left(\frac{\omega_c}{s + \omega_c}\right) q_i(s) \tag{A4}$$

$$P_{DSOGI}(s) = \left(\frac{s^2 + (2\omega_o)^2}{s^2 + 2(2\omega_o) \cdot s + (2\omega_o)^2}\right) p_i(s) \tag{A5}$$

$$Q_{DSOGI}(s) = \left(\frac{s^2 + (2\omega_o)^2}{s^2 + 2(2\omega_o) \cdot s + (2\omega_o)^2}\right) q_i(s) \tag{A6}$$

where $\omega_c = 2\pi f_c$ is the cut-off frequency in rad/s of the LPFs used in the droop- and advanced-based methods. The instantaneous powers, $p_i(t)$ and $q_i(t)$, are derived from the product between $i_o(t)$ and $v_o(t)$ and $v_{o\perp}(t)$, respectively. The transfer functions for $v_o(t)$ and $v_{o\perp}(t)$ correspond to Equations (15)

and (16) in all the methods, since a SOGI was used to achieve these signals. In addition, in the case of the current, the fundamental component $i_{OF}$ in the DSOGI method has the following transfer function:

$$i_{OF}(s) = \left( \frac{2\xi_c \omega_o \cdot s}{s^2 + 2\xi_c \omega_o \cdot s + \omega_c^2} \right)^2 i_o(s) \tag{A7}$$

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
