# Peer review of "A Power Calculation Algorithm for Single-Phase Droop-Operated-Inverters Considering Linear and Nonlinear Loads HIL-Assessed"

_electronics, doi:10.3390/electronics8111366_

Round 1

Reviewer 1 Report

Please reflect following comments for revision.

Please add technical details (such as transfer functions) and description of the proposed P-Q calculation scheme in the section 2.4. For the simulation and experimental results, please add results of other physical variable such as voltage, current, reactive power (Q). Please indicate command (or reference) signals and measured signals. Input signal for P-Q calculations are voltage and current. The voltage and current are also output (results) signals of P-Q command. In the paper, correlation of voltage, current, P, Q are not clearly presented. For experimental results, it would be clearer to show improvement if results are overlaid each other.     

Author Response

Thank you very much for your comments, below there is the detailed answer to them:

Comment: Calculation scheme in the section 2.4. For the simulation and experimental results, please add results of other physical variable such as voltage, current, reactive power (Q).

Answer: New figures and results are added to the paper. In particular, you can find the VSI output voltage, asymmetrical nonlinear load current and extracted fundamental current component in Figure 18. Regarding Q the value for this load is small, of 20W only, and we decide not to show it because can lead to misunderstanding in the design and comparison of the different methods. The ripple for Q is different for the different methods. The design should be focused on P or in Q for achieving the same power ripple. Equal ripple can be obtained for one of this powers, but not for both. So, we decide to choose P, since it’s the higher power rated.

Comment: Please indicate command (or reference) signals and measured signals. Input signal for P-Q calculations are voltage and current. The voltage and current are also output (results) signals of P-Q command. In the paper, correlation of voltage, current, P, Q are not clearly presented.

Answer: You are right, but we don’t have command or references. The input to the P-Q blocks are the VSI output voltage, vo(t), and output current, io(t). The P and Q are derved after the calculations and filtering over the instantaneous powers, pi(t) and qi(t). We don’t show these waveforms because they are too noise (high distortion), due to the nonlinear current waveform of the output current. And this happens for the previous methods. So, only the filtered powers are shown. In our proposal, we show the fundamental component in Figure 18. We use the DSOGI to do a strong filtering. The waveform obtained after are almost linear and the SOGI1 and SOGI2 of Figure 13 remove the double frequency component. The figures associated with this are shown in previous papers when dealing only with linear loads.

Reviewer 2 Report

Dear Authors, please look at the requests in the attached file

Author Response

Thank you very much for you detailed revision, comments and suggestions that help us to improve the paper. We have introduced all them. Moreover, we have done an extra revision of the paper and introduced new figures to better explain the ideas. Look at the new version that you will find that the most part has been changed and is highlighted in yellow.

Reviewer 3 Report

This paper is very similar to the ref. 29. The authors have just added some HIL results in this paper in comparing to the ref. 29.
The structure of the paper must be improved. There are many long and hard to understand sentences in the text. The section also must be rearranged.

Best regards

Ehsan JAMSHIDPOUR

Author Response

Answer: Thank you very much to you comment. This paper is similar to ref. 29 because we presented the idea in a congress in a 6 pages long paper with only simulations results. Now the paper includes HIL results but also experimental results with a single-phase inverter Danfos from the Lab. We achieved this results this August. The paper now included more figures and is better tuned and explained in consonance with the HIL and experimental test. If you take a look to the revised paper now, you will see that it has been completely re-written, I took a special care to remove, or split, long sentences. And, the paper has been rearranged as you suggested. Now it is more clear and easy to follow.

Round 2

Reviewer 1 Report

Some of my comments from the first review were ignored.

Authors are sincerely asked to answer why comments were ignored.

The first sentence of my first review is as follows.

"Please add technical details (such as transfer functions) and description of the proposed P-Q calculation scheme in the section 2.4."

Besides section 2.4, other sections are review and explanation of existing materials and HILS and experimental results. Section 2.4 is the main part of this paper. Therefore, adding technical details and description are mandatory.

My last comment was to overlay experimental results in Fig. 19,20, and 21 for clear comparison of the results. This comments was ignored. Please explain the reason. 

Reviewer 3 Report

Dear Authors, 

Thank you for the modifications, the paper is really improved. 

Best regards 

Author Response

Thanks for your answer. This is the answer to the second round of revision. We are thankful for your effort. There is no comments in the review.

Best regards,

Jordi El Mariachet Carreño

Round 3

Reviewer 1 Report

none.